# Validation and Concordance Analysis of a New Lateral Flow Assay for Detection of *Histoplasma* Antigen in Urine

**DOI:** 10.3390/jof7100799

**Published:** 2021-09-24

**Authors:** Diego H. Cáceres, Beatriz L. Gómez, Ángela M. Tobón, Melissa Minderman, Nicole Bridges, Tom Chiller, Mark D. Lindsley

**Affiliations:** 1Centers for Disease Control and Prevention CDC, Atlanta, GA 30329, USA; tnc3@cdc.gov (T.C.); mil6@cdc.gov (M.D.L.); 2Center of Expertise in Mycology Radboudumc/CWZ, 6525 Nijmegen, The Netherlands; 3Studies in Translational Microbiology and Emerging Diseases (MICROS Research Group), School of Medicine and Health Sciences, Universidad del Rosario, Bogota 111221, Colombia; beatrizlgomez@hotmail.com; 4Hospital La Maria, Medellín 050040, Colombia; angelamtobon@hotmail.com; 5Instituto Colombiano de Medicina Tropical, Universidad CES, Medellín 050021, Colombia; 6MiraVista Diagnostics, Indianapolis, IN 46241, USA; MMinderman@miravistalabs.com (M.M.); nbridges@miravistalabs.com (N.B.)

**Keywords:** histoplasmosis, *Histoplasma*, antigen, HIV, AIDS

## Abstract

Histoplasmosis is a major cause of mortality in people living with HIV (PLHIV). Rapid methods to diagnose *Histoplasma capsulatum* disease could dramatically decrease the time to initiate treatment, resulting in reduced mortality. The aim of this study was to validate a MiraVista^®^ Diagnostics (MVD) *Histoplasma* urine antigen lateral flow assay (MVD LFA) for the detection of *H. capsulatum* antigen (Ag) in urine and compare this LFA against the MVista^®^ *Histoplasma* Ag quantitative enzyme immunoassays (MVD EIA). We assessed the MVD LFA using a standardized reference panel of urine specimens from Colombia. We tested 100 urine specimens, 26 from PLHIV diagnosed with histoplasmosis, 42 from PLHIV with other infectious diseases, and 32 from non-HIV infected persons without histoplasmosis. Sensitivity and specificity of the MVD LFA was 96%, compared with 96% sensitivity and 77% specificity of the MVD EIA. Concordance analysis between MVD LFA and the MVD EIA displayed an 84% agreement, and a Kappa of 0.656. The MVD LFA evaluated in this study has several advantages, including a turnaround time for results of approximately 40 min, no need for complex laboratory infrastructure or highly trained laboratory personnel, use of urine specimens, and ease of performing.

## 1. Introduction

Histoplasmosis is a disease caused by *Histoplasma capsulatum*, a thermally dimorphic fungus. Infection is most frequently diagnosed in the Americas, but histoplasmosis has been reported worldwide [1,2]. *H. capsulatum* is associated with soil contaminated with bird excreta and bat guano, and infection occurs when the host inhales the infectious fungal microconidia and mycelial fragments after disruption or aeration of the contaminated soil [1,2]. Histoplasmosis is primarily a pulmonary disease but may secondarily spread to other organs, especially those of the reticuloendothelial system [3,4]. 

Disseminated histoplasmosis (DH) is the most frequent clinical presentation of histoplasmosis in people living with HIV/AIDS (PLHIV). This infection can disseminate to skin, bone, adrenal glands, the gastrointestinal tract, and the central nervous system [3,4]. Symptoms of DH are often nonspecific, and among patients with advanced HIV disease, they may be similar to those of other infectious diseases, especially tuberculosis, complicating diagnosis and treatment [5,6,7,8]. In Colombia and other countries of the Americas where histoplasmosis is endemic and HIV is highly prevalent, late detection of histoplasmosis, resulting in delayed treatment, is a major cause of mortality (approximately 30% in PLHIV) [3,9,10]. 

Conventional laboratory methods used for diagnosis of histoplasmosis, such as culture and histopathology, pose many challenges, including the need for complex laboratory infrastructure and staff with mycology training, a turnaround time of up to several weeks, and variable assay sensitivity (~60%) [11]. Diagnosis using serologic methods is further complicated by a reduction in antibody test sensitivity (38–70%) when performed in immunocompromised persons [11]. Detection of circulating antigen has been described as the best option to diagnose DH in PLHIV [11,12]. 

The first assay for the detection of *Histoplasma* antigen (Ag) in urine was developed in 1986; changes and improvements have been made to the assay, and the current version of the test has a high sensitivity, over 95% [13,14,15]. MiraVista^®^ Diagnostics (MVD) (Indianapolis, IN, USA) offered the service of *Histoplasma* antigen detection in their reference laboratory, using an enzyme immunoassays (EIA) [13]. In 2015, IMMY (Norman, OK, USA) developed the Clarus *Histoplasma* GM EIA, which is commercially available for individual lab use. Reports on the use of this assay have also described high sensitivity and specificity for the diagnosis of histoplasmosis in PLHIV [16,17,18,19,20,21,22,23,24,25]. 

Recently, a lateral flow assay (LFA) for the detection of *Histoplasma* Ag in urine has been developed by MVD. The MVD LFA is designed to be rapid, simple to perform, and portable, potentially making testing available to resource-challenged countries, where it is needed most. This LFA was recently validated in a multicenter cohort of PLHIV from Mexico, where it showed high performance, 92% accuracy, to diagnose histoplasmosis [17]. The aim of the current study was to evaluate the analytical performance of the MVD LFA assay and compare it to the MiraVista^®^ *Histoplasma* Ag quantitative enzyme immunoassays (MVD EIA), using a standard reference collection of urine samples from Colombian PLHIV.

## 2. Materials and Methods

***Study specimens:*** For use in this study, a standardized reference panel of 100 remnant urine specimens were selected from specimens obtained during a study of histoplasmosis in PLHIV conducted at the Hospital La María in Medellín, Colombia [26]. Details of the patient enrollment criteria have been published [16]. All patients from this study [26] were tested using multiple laboratory assays for fungal and mycobacteria infections. These testing was performed at the Medical and Experimental Mycology Unit at Corporación para Investigaciones Biológicas (CIB), a Colombian laboratory specialized in medical mycology, certified by the International Organization for Standardization (ISO) 9001:2015 and the Colombian regulatory agencies for medical services. As part of the CIB clinical laboratory protocol, patient specimens including blood, tissues, sterile body fluids and respiratory specimens were tested by culture and microscopy for fungal, mycobacteria, and other bacterial infections. Each specimen received a direct exam for fungal and mycobacterial pathogens using KOH 10%, Wright stain and Nigrosin stain (fungi) and Auramine-Rhodamine stain (mycobacteria). All tissue, body fluid and respiratory specimens were cultured on Sabouraud agar with antibiotics, Mycosel and niger seed agar, for fungal detection, and on Löwenstein–Jensen medium and the BD BACTEC™ MGIT™ automated mycobacterial detection system. Blood cultures were performed using Myco/F lytic culture vials, incubated on the BD BACTEC^TM^ automated system for a total of 42 days. Additionally, immunodiagnostics assays were performed, including immunodiffusion and complement fixation for fungal pathogens, *Histoplasma* spp, *Paracoccidioides* spp., and *Aspergillus* spp, and *Histoplasma* antigen detection [16,26]. Serological testing was confirmed by the Centers for Disease Control and Prevention (CDC), Mycotic Diseases Branch (MDB) reference laboratory, using Clinical Laboratory Improvement Amendment (CLIA)-approved assays. A final patient diagnosis was then established based on laboratory results and review of the patient’s clinical records [27]. Samples were then stored at –80 ºC until the time of analysis for this study. 

For the current study, a total of 100 urine specimens were used (Figure 1): 26 urine specimens from patients diagnosed with DH (24 proven and 2 probable); 42 urine specimens from non-histoplasmosis patients who had a final diagnosis with another infectious disease determined by culture or another gold standard technique, as described above. These included patients diagnosed with paracoccidioidomycosis (PCM) (*n* = 2), cryptococcosis (*n* = 10), candidiasis (*n* = 1), aspergillosis (*n* = 1), pulmonary pneumocystosis (*n* = 3), tuberculosis (*n* = 23) and toxoplasmosis (*n* = 2); and 32 from non-HIV infected and non-suspected to have histoplasmosis persons, residing in a region endemic for histoplasmosis, were also included in the evaluation (Figure 1). All specimens were blinded prior to testing. 

***Lateral flow assay (LFA) and enzyme immunoassay (EIA) testing for the detection of Histoplasma antigen*:** An LFA for the detection of the *Histoplasma* antigen was provided by MiraVista^®^ Diagnostics (Indianapolis, IN, USA) and performed at CDC laboratories. The MVD LFA is a dipstick sandwich immunochromatographic assay using a rabbit polyclonal antibody (PoAb) that recognizes the *H. capsulatum* galactomannan antigen. Urine specimens were tested according to the manufacturer’s instructions. Positive results were interpreted as the presence of two lines, a test line and a control line. A negative result was interpreted as the presence of the control line alone. The presence of no lines, or a test line in the absence of a control line, was interpreted as an invalid result. The presence of a line was visually performed by two independent readers. 

The MiraVista^®^ *Histoplasma* Ag quantitative EIA (MVD EIA) was performed at the MiraVista^®^ Diagnostics laboratory using standard quality controls and assessment methods. Urine samples (250 µL) were blinded and shipped overnight, frozen, from Atlanta, GA, USA, to Indianapolis, IN, USA. The MVD EIA is a quantitative EIA that provides results within the linear range of 0.2 to 20.0 ng/mL [28]. Results where no *Histoplasma* Ag was detected were classified as negative. Positive results were reported using the quantitative value of the antigen concentration. Results greater than 20 ng/mL were outside the linear range and designated as Ag concentrations above the limit of quantification (ALQ), and specimens with *Histoplasma* antigen results less than 0.2 ng/mL were determined to have Ag concentrations below the limit of quantification (BLQ). Both BLQ and ALQ were considered positive results. 

***Statistical analysis:*** Calculation of the analytical performance of the test was done using 2 × 2 tables comparing MVD LFA and MVD EIA results versus the patient’s diagnosis; the test sensitivity, specificity, accuracy, and positive and negative predictive values were calculated, with their respective 95% confidence intervals (CI). A concordance analysis between the MVD LFA and the MVD EIA was performed; this analysis includes determining the proportion of results in agreement between MVD LFA and MVD EIA, and the Kappa value with its respective CI [29]. Analyses were conducted using STATA 11 software and EPIDAT 4.2. 

***Ethics:*** This study was performed according to the terms agreed upon and with the full approval of the ethical committees of the CDC, CIB, and Hospital La María in Medellin, Colombia (Protocol 4250). All patients enrolled in this study signed an informed consent form designed in collaboration with the ethical committee of the CIB. All clinical information from the participants in the study was anonymized in a database using an alphanumerical code.

## 3. Results

**Evaluation of the analytical performance of the MVD *Histoplasma* urinary Ag LFA.** The results of the MVD LFA correctly classified as positive 25 of the 26 urine samples (96% sensitivity, CI 80–100%) from patients with histoplasmosis (Table 1). One false negative LFA result was observed from a patient who was diagnosed with culture-confirmed histoplasmosis (Table 1).

In specimens from non-histoplasmosis patients, negative LFA results were correctly reported in 71 of 74 (96% specificity, CI 89–99%) urine specimens tested (Table 1). Three false positive results were observed, including two urine samples from patients with PCM (Table 2, specimens 1 and 2), and one in a sample from a patient with cryptococcosis without clinical, laboratory or epidemiological evidence of histoplasmosis (Table 2, specimen 3). The MVD LFA displayed an accuracy of 96% (CI 90–99) for the diagnosis of DH. 

**Evaluation of the analytical performance of the MVD *Histoplasma* urinary Ag quantitative EIA.** Using the MVD EIA, *Histoplasma* antigen was detected in the urine of 25 of the 26 histoplasmosis patients, yielding a sensitivity of 96% (CI 80–100%). As observed with the MVD LFA, one false negative EIA result was observed from a patient who was diagnosed with culture-confirmed histoplasmosis (Table 1 and Table 2 specimen 19). 

In the 74 specimens from patients without histoplasmosis, negative EIA results were correctly reported in 57 specimens, resulting in a specificity of 77% (CI 66–86%). Urine specimens from seventeen patients displayed culture discrepant results; all were from non-histoplasmosis patients: PCM (*n* = 2) (Table 2, specimens 1 and 2), tuberculosis (*n* = 7) (Table 2, specimens 5 to 11), *Pneumocystis* pneumonia (*n* = 2) (Table 2, specimens 12 and 13), cryptococcosis (*n* = 1; a different patient than the MVD LFA false positive result) (Table 2, specimen 4), toxoplasmosis (*n* = 1) (Table 2, specimen 14) and four non-HIV infected persons who were residents in an endemic area for histoplasmosis (Table 2, specimens 15 to 18). None of the subjects with false positive *Histoplasma* antigen test results presented any laboratory, clinical, or epidemiological evidence of histoplasmosis. The MVD EIA displayed an accuracy of 82% (CI 73–89%) for the diagnosis of DH.

**Performance of concordance analysis between the MVD *Histoplasma* Urine Ag LFA and EIA.** A total of 84 (84%) results were in agreement between the MVD LFA and MVD EIA. Concordance between both assays showed good agreement with a kappa index of 0.66 (CI 0.51–0.80) (Table 1).

## 4. Discussion

Diagnosis of histoplasmosis is challenging. Invasive procedures may be required for specimen collection, and conventional laboratory tests, such as culture and special histopathologic stains, requiring specialized training, and often being time consuming, thereby extending the time to diagnosis [2,4,11]. This study describes a newly developed, rapid, highly sensitive, and commercially available *Histoplasma* urinary Ag LFA. This test offers a particularly promising approach to prompt diagnosis of histoplasmosis in resource limited settings using urine and providing results within minutes after specimen collection.

Specimens used in this study are from a unique, well characterized panel of urine samples from patients with and without histoplasmosis, all culture-confirmed. In patients with non-*Histoplasma* diseases, where culture was not feasible, serology and special stains were used for pathogen identification and diagnosis. These specimens have been used in multiple studies of other antigen detection systems for diagnosing histoplasmosis [16,26,30]. 

Excellent sensitivity was observed with both the MVD LFA and MVD EIA (96% for both). The one DH patient who tested negative by both assays had received trimethoprim/sulfamethoxazole (TMP/SMX) as prophylaxis for pulmonary pneumocystosis. TMP/SMX has been used for the treatment of endemic mycoses [31], which may have lowered the fungal burden in this patient, reducing the antigen concentration below the test’s limit of detection [32]. In addition, this specimen was previously tested using other antigen assays, including the MVD EIA in this study, producing similar false-negative results.

The performance of the MVD LFA and the MVD EIA differed in relation to assay specificity: 96% vs 77%, respectively. The MVD LFA exhibited three discrepant results compared with culture. Two of the discrepant results involved cross-reactions observed in patients with paracoccidioidomycosis. These patients were culture-negative for *Histoplasma* and were also seronegative by *Histoplasma* antibody assays. False positive reactions with specimens from these *Paracoccidioides* patients have also been observed in other studies that evaluated *H. capsulatum* antigen assays and most likely represent true cross-reactivity in patients with PCM [16,26,30,33]. While this cross-reactivity is important to note, antifungal treatment is similar and PCM is observed much less frequently than histoplasmosis in PLHIV [2]. 

The third discrepant result, observed using the MVD LFA, corresponded to a false positive result observed in a specimen from a patient with confirmed cryptococcosis, without evidence of histoplasmosis. Interestingly, this patient has tested negative for urinary *Histoplasma* antigen using three different antigen detection assays, including the MVD EIA evaluated in this study. With MVD EIA, 17 false positive results were observed. Fifteen were from patients without epidemiological or clinical evidence of histoplasmosis. The other two were the patients diagnosed with PCM where cross-reactivity was also observed with the MVD LFA. Antigen concentrations in these false positive patients were considered positive, because they were below the limit of quantification (*n* = 4), had Ag concentrations ranging between 0.23 to 1.0 ng/mL (*n* = 7), or had concentrations between 1 and 2 ng/mL (*n* = 3). In prior studies, greater specificity results have been reported using MVD EIA [14,15,34]. While specific patient control groups, such as those infected with other endemic molds or *Aspergillus*, are defined, for others, described as non-histoplasmosis or non-fungal patients, a final diagnosis was not provided [14,15,34]. In the current study, false positive results were observed in patients diagnosed with tuberculosis, cryptococcosis, and pulmonary pneumocystosis. These diseases can mimic histoplasmosis and are important to differentiate from each other in order to provide appropriate therapy.

Overall, the MVD LFA and MVD EIA have good agreement, but the MVD LFA was more accurate than the MVD EIA; 96% (CI 90–99%) vs 82% (CI 73–89%). In a recent prospective study from Mexico involving 104 urine samples from patients with histoplasmosis and 287 from patients without, the sensitivity of the MVD LFA was 90% (CI 83–95%), specificity was 92% (CI 89–95), and accuracy was 92% (89–94) [17]. In the current study, the MVD LFA displayed a sensitivity, specificity, and accuracy of 96% for each.

Limitations of this study are principally related to the sample size and the lack of specimens from patients with other mycoses, including coccidiomycosis, blastomycosis and talaromycosis, some of which have displayed cross-reactivity in other *Histoplasma* antigen assays [33]. False positive reactions with the MVD EIA were observed in this study. Undetected dual infection with *Histoplasma* in these patients might explain these results; however, the extensive laboratory and clinical workup of these patients, as described in the methods, makes this unlikely. Furthermore, these same specimens have been used to evaluate similar assays without false positive interference [16,26,30]. Finally, the MVD LFA was not evaluated for the diagnosis of non-disseminated clinical forms of histoplasmosis.

## 5. Conclusions

In conclusion, the MVD LFA is highly sensitive and specific for detection of histoplasmosis in PLHIV. The MVD LFA can be performed in less than an hour without complex laboratory infrastructure or highly trained laboratory personnel and employs urine, a non-invasive, easily obtained clinical specimen. The results and considerations presented in this study lead us to conclude that the MVD LFA is a promising tool for point of care testing of people suspected to have histoplasmosis.

## Figures and Tables

**Figure 1 jof-07-00799-f001:**
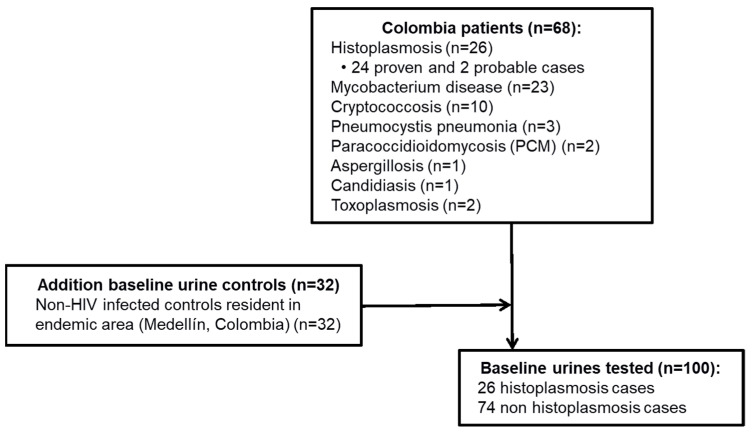
Study subjects and urine specimens analyzed during the validation of the commercial MVD *Histoplasma* Urine Ag lateral flow assay.

**Table 1 jof-07-00799-t001:** Analytical performance of the MVD *Histoplasma* urinary Ag LFA and MVD EIA.

		MVD LFA				MVD EIA		
			Culture				Culture	
	**LFA**		+	−	**EIA**		+	−
		+	25	3		+	25	17
		−	1	71		−	1	57
		% (CI 95%)						
**Sensitivity**		96% (80–100)				96% (80–100)		
**Specificity**		96% (89–99)				77% (66–86)		
**Accuracy**		96% (90–99)				82% (73–89)		
**Positive predictive value**		89% (73–96)				60% (49–69)		
**Negative predictive value**		99% (91–100)				98% (89–100)		
**Kappa**		0.656 (0.509–0.803)						

(LFA) Lateral flow assay; (EIA) Enzyme Immunoassay; (95% CI confidence interval); (+) positive result; (−) negative result.

**Table 2 jof-07-00799-t002:** Characteristics of patients’ specimens with results discrepant between culture/diagnosis and the MVD *Histoplasma* urinary Ag LFA and EIA.

#	Patient Diagnosis	LFA Result	EIA Result (ng/mL)	Laboratory Diagnosis	Comments
** *Discrepant results against fungal negative culture* **					
1	PCM	**+**	ALQ (+)	PCM proven by culture	Positive *Paracoccidioides* serology, HIV negative
2	PCM	**+**	5.7 (+)	PCM proven by culture.	Positive *Paracoccidioides* serology, HIV negative
3	Cryptococcosis	**+**	−	Culture/microscopy in CSF and lymph node. (+) Ag by latex in CSF	Negative fungal serology, 4 CD4 cells/mm^3^, (−) *Histoplasma* culture,
4	Cryptococcosis	−	0.88 (+)	*Cryptococcosis* proven by culture/microscopy in lung tissue	Negative fungal serology, (−) *Histoplasma* culture,
5	Tuberculosis	−	2.4 (+)	Tuberculosis proven by positive AFB smear	Negative fungal serology, 186 CD4 cells/mm^3^, (−) *Histoplasma* culture
6	Tuberculosis	−	1.9 (+)	Tuberculosis proven by culture and/microscopy in BAL	Negative fungal serology, 44 CD4 cells/mm^3^, (−) *Histoplasma* culture
7	Tuberculosis	−	1.2 (+)	Tuberculosis proven by positive AFB smear	Negative fungal serology, 241 CD4 cells/mm^3^, (−) *Histoplasma* culture
8	Tuberculosis	−	0.8 (+)	Tuberculosis proven by culture/microscopy in BAL	Negative fungal serology, (−) *Histoplasma* culture, (−) by four of five *Histoplasma* Ag assays
9	Tuberculosis	−	0.4 (+)	Tuberculosis proven by blood culture	Negative fungal serology, (−) *Histoplasma* culture, Patient deceased
10	Tuberculosis	−	BLQ (+)	Tuberculosis proven by culture and/microscopy in sputum and lymph node	Negative fungal serology, (−) *Histoplasma* culture
11	Tuberculosis	−	BLQ (+)	Tuberculosis proven by culture and/microscopy in sputum	Negative fungal serology, 242 CD4 cells/mm^3^, (−) *Histoplasma* culture
12	PjP	−	0.7 (+)	PjP proven by special staining	Negative fungal serology, 15 CD4 cells/mm^3^, (−) *Histoplasma* culture
13	PjP	−	BLQ (+)	PjP proven by special staining	Negative fungal serology, (−) *Histoplasma* culture,
14	Toxoplasmosis	−	BLQ (+)	Toxoplasma proven by serology	Negative fungal serology, (−) *Histoplasma* culture, patient on TB treatment
15	Non-HIV	−	1.5 (+)	Asymptomatic; resident of region endemic for histoplasmosis	Negative fungal serology
16	Non-HIV	−	0.4 (+)	Asymptomatic; resident of region endemic for histoplasmosis	Negative fungal serology
17	Non-HIV	−	0.3 (+)	Asymptomatic; resident of region endemic for histoplasmosis	Negative fungal serology
18	Non-HIV	−	0.3 (+)	Asymptomatic; resident of region endemic for histoplasmosis	Negative fungal serology
** *False negative* **					
19	DH	−	−	DH proven by culture	(+) *Histoplasma* culture in peritoneum, negative fungal serology, 570 CD4 cells/mm^3^, HIV viral load < 40 copies,

(+) positive result; (−) negative result; (#) number; (Ag) antigen; (AFB) acid-fast bacillus; (CSF) cerebrospinal fluid; (BAL) bronchoalveolar lavage; (DH) disseminated histoplasmosis; (PCM) paracoccidioidomycosis; (PjP) *Pneumocystis* pneumonia. All infectious diseases were co-infections with HIV in these patient cohorts. MVD EIA result interpretation: (BLQ) below limit of quantification; (ALQ) above limit of quantification.

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
