# Peer review of "Validation and Concordance Analysis of a New Lateral Flow Assay for Detection of Histoplasma Antigen in Urine"

_jof, 2021, doi:10.3390/jof7100799_

Round 1
Reviewer 1 Report
Second paragraph on page 2: Since the Mira Vista assay was developed, IMMY also developed a plate ELISA assay for the detection to Histo Unitary antigen. Many reference labs use this assay instead of the MiraVista lab. The Miravista assay has been optimized to test also serum, BAL and even CSF. Of note OID has also developed a lateral flow assay that is being submitted to FDA.TABLE 1 is nice. Table 2 is an eye opener. This is a really special specimen set to be used for this validation. The number of false positives by the ELISA surprises me. There are several issues with the panel as commented upon by the authors in the first paragraph on page 7. The most substanitive issue is that these patients all had disseminated disease and thus are more likely to be positive using any Histo Uag. Second is that there are no Blastomyces infections were tested. Knowing if the LAF will be cross-reactive with Blastomyces, Coccidioides and other endemic mycoses we know that the ELISA detects cases of Blastomyces, which in our hands have been some of the highest histo urinary antigen we have seen using the MiraVista ELISA assay. I think that it is an oversight that no cases were testing for this study to supplement the cases from Colombia.
Aside from these issues, I think this manuscript is well written ad very important. I would recommend obtaining some samples reflecting cases of other endemic mycoses and test these to determine the degree of cross reactivity seen with these other infective agents. If this assay is really to be of use in the US. I think that additional study looking at the sensitivity of the assay in detecting disease in patients who do not have disseminated disease would also be important, but probably not within the scope of this manuscript.
Author Response
Reviewer 1:
Second paragraph on page 2: Since the Mira Vista assay was developed, IMMY also developed a plate ELISA assay for the detection to Histo Unitary antigen. Many reference labs use this assay instead of the MiraVista lab. The Miravista assay has been optimized to test also serum, BAL and even CSF. Of note OID has also developed a lateral flow assay that is being submitted to FDA.TABLE 1 is nice. Table 2 is an eye opener. This is a really special specimen set to be used for this validation. The number of false positives by the ELISA surprises me. There are several issues with the panel as commented upon by the authors in the first paragraph on page 7. The most substanitive issue is that these patients all had disseminated disease and thus are more likely to be positive using any Histo Uag. Second is that there are no Blastomyces infections were tested. Knowing if the LAF will be cross-reactive with Blastomyces, Coccidioides and other endemic mycoses we know that the ELISA detects cases of Blastomyces, which in our hands have been some of the highest histo urinary antigen we have seen using the MiraVista ELISA assay. I think that it is an oversight that no cases were testing for this study to supplement the cases from Colombia.
Aside from these issues, I think this manuscript is well written ad very important. I would recommend obtaining some samples reflecting cases of other endemic mycoses and test these to determine the degree of cross reactivity seen with these other infective agents. If this assay is really to be of use in the US. I think that additional study looking at the sensitivity of the assay in detecting disease in patients who do not have disseminated disease would also be important, but probably not within the scope of this manuscript.
- Thank you for your comments. We extended in the introduction to include the description of other antigen test available for the clinical diagnosis of histoplasmosis. We also added references describing the implementation of IMMY EIA. We know about the OIDx LFA, we did not include this assay as there are no data, at this time, available supporting the validation of this LFA.
We agree, the lack of clinical specimens from patients with coccidiomycosis, blastomycosis and talaromycosis it is a limitation of this study. This is described as a limitation in the discussion (page 7, lines: 242 and 243). These diseases are not endemic in Colombia, the reason why no cases were identified during the cohort collection.
Reviewer 2 Report
Well written paper!
- Page 3: How was the LFA line read? By vision? by two examiners?
- page 3: For the MVD EIA, I guess that there is for each run a positive, a negative control, and a standard curve?
- finally: I think that these results are great. I hope that this test will be commercially available at an affordable price, as it could improve substantially the diagnosis and treatment of the disease in endemic regions and potentially help to elucidate the epidemiology in Africa.
Author Response
Reviewer 2:
Page 3: How was the LFA line read? By vision? by two examiners?
- Thanks, we clarified LFA interpretation in the material and methods section (page 3, lines: 122 and 123).
page 3: For the MVD EIA, I guess that there is for each run a positive, a negative control, and a standard curve?
- A statement was added to the methods section regarding MVD EIA methods to state, “The MiraVista® Histoplasma Ag quantitative EIA (MVD EIA) was performed at the MiraVista® Diagnostics laboratory using standard quality controls and assessment methods.” The added wording to the test is underlined.
finally: I think that these results are great. I hope that this test will be commercially available at an affordable price, as it could improve substantially the diagnosis and treatment of the disease in endemic regions and potentially help to elucidate the epidemiology in Africa.
R. We hope the same.